# Stereotypies in the Autism Spectrum Disorder: Can We Rely on an Ethological Model?

**DOI:** 10.3390/brainsci11060762

**Published:** 2021-06-08

**Authors:** Roberto Keller, Tatiana Costa, Daniele Imperiale, Annamaria Bianco, Elisa Rondini, Angela Hassiotis, Marco O. Bertelli

**Affiliations:** 1Adult Autism Centre, Mental Health Department, ASL Città di Torino, 10138 Turin, Italy; rokel2003@libero.it (R.K.); costatatiana@gmail.com (T.C.); 2Neurology Unit, Maria Vittoria Hospital, ASL Città di Torino, 10144 Turin, Italy; daniele.imperiale@aslcittaditorino.it; 3CREA (Research and Clinical Centre), San Sebastiano Foundation, Misericordia di Firenze, 50142 Florence, Italy; abianco@crea-sansebastiano.org (A.B.); elisa.rondini@yahoo.it (E.R.); 4Division of Psychiatry, University College London, London W1T 7NF, UK; a.hassiotis@ucl.ac.uk

**Keywords:** autism spectrum disorder, stereotypies, repetitive behaviours, restricted behaviour, ethological model

## Abstract

Background: Stereotypic behaviour can be defined as a clear behavioural pattern where a specific function or target cannot be identified, although it delays on time. Nonetheless, repetitive and stereotypical behaviours play a key role in both animal and human behaviour. Similar behaviours are observed across species, in typical human developmental phases, and in some neuropsychiatric conditions, such as Autism Spectrum Disorder (ASD) and Intellectual Disability. This evidence led to the spread of animal models of repetitive behaviours to better understand the neurobiological mechanisms underlying these dysfunctional behaviours and to gain better insight into their role and origin within ASD and other disorders. This, in turn, could lead to new treatments of those disorders in humans. Method: This paper maps the literature on repetitive behaviours in animal models of ASD, in order to improve understanding of stereotypies in persons with ASD in terms of characterization, pathophysiology, genomic and anatomical factors. Results: Literature mapping confirmed that phylogenic approach and animal models may help to improve understanding and differentiation of stereotypies in ASD. Some repetitive behaviours appear to be interconnected and mediated by common genomic and anatomical factors across species, mainly by alterations of basal ganglia circuitry. A new distinction between stereotypies and autotypies should be considered. Conclusions: Phylogenic approach and studies on animal models may support clinical issues related to stereotypies in persons with ASD and provide new insights in classification, pathogenesis, and management.

## 1. Introduction

Autism spectrum disorder (ASD) is a neurodevelopmental disorder characterized by persistent deficits in social communication and social interaction across multiple contexts, and restricted, repetitive patterns of behaviour, interests, or activities [1,2].

The term ‘restrictive and repetitive behaviour’ (RRB) and its common alternative ‘abnormal repetitive behaviour’ (ARB) describe a wide range of behaviours, which share three common characteristics [3]: (1) the behaviour is displayed with high frequency of repetition; (2) it is performed in an invariant way; (3) the behaviour’s manifestation is inappropriate or odd.

In ASD, RRBs are better defined by the presence of at least two of the following groups of symptoms: (i) stereotyped or repetitive motor movements, use of objects, or speech; (ii) insistence on sameness, inflexible adherence to routines, or ritualized patterns of verbal or nonverbal behaviour; (iii) highly restricted, fixated interests that are abnormal in intensity or focus; and (iv) hyper- or hypo-reactivity to sensory input or unusual interest in sensory aspects of the environment [1,2].

This broad range of behaviours has been conceptualized in two clusters: (1) “lower-order” motor actions (stereotyped movements, repetitive manipulation of objects and repetitive forms of self-injurious behaviour) characterized by repetition of movement, and (2) “higher-order” behaviours (compulsions, rituals, insistence on sameness, and circumscribed interests) that have a distinct cognitive component. The latter are characterized by adherence to some rule or mental set [4,5]. This categorization has been empirically supported by factor analyses, using relevant items from the Autism Diagnostic Interview Revised (ADI-R), which represents a standardized, semi-structured caregiver interview that is considered to be a “gold standard” measure in the assessment of a range of behaviours consistent with diagnoses of ASD. Such factors have been labelled as repetitive sensory motor behaviour and resistance to change or insistence on sameness [6,7].

Stereotypies are defined as repetitive and topographically invariant acts, without a clearly established purpose or function [8]. Examples include hand flapping, body-rocking, head rolling, etc. [9].

RRB are commonly observed in a variety of developmental, psychiatric and neurological disorders other than ASD, including Rett syndrome, Fragile X syndrome, intellectual disability, schizophrenia, Parkinson disease, dementia, Tourette syndrome, and obsessive–compulsive disorder, which can lead to issues with differential diagnosis or comorbidity with ASD [10,11,12]. For example, certain forms of ASD and obsessive compulsive disorder may share a number of clinical features related to RRB that make it extremely difficult to distinguish the two conditions and lead to erroneous overdiagnosis of comorbidity.

In spite of the relevant significance of repetitive behaviours in daily clinical practice with persons with ASD, devoted literature is relatively scarce with respect to plenty of studies on social and communication deficits. On the contrary, a huge amount of research on stereotypies and repetitive behaviour was carried out on animal models, because motor stereotypies are easier to model in animals, and higher-order repetitive behaviours in animals were thought to result from secondary neuropathological changes [5,13,14].

Since ASD is characterized by the co-occurrence of “lower-order” and “higher-order” repetitive behaviours [11], it is important that relevant animal models include attempts to model both motor and cognitive features of repetitive behaviours [15].

Stereotypies are a major source of stress for parents, resulting in considerable accommodation by the family and negative impact on academic achievement [16]. Nonetheless, treatment options for ARB are limited [17]. To date, a wide range of psychotropic medications [e.g., antipsychotics, selective serotonin-reuptake inhibitors (SSRIs) and serotonin and norepinephrine reuptake inhibitors (SNRIs)] have been used, but there is no established drug-based treatment. Evidence on the efficacy of these medications is inconsistent, and their prescription is limited by the possibility of long-term adverse side effects [18,19,20].

Some compounds, such as clomipramine, fluvoxamine, fluoxetine, sertraline, citalopram and venlafaxine were found to have some efficacy, but they are rarely prescribed because of lack of knowledge on safety and tolerability [20]. There are few pharmacological interventions with established efficacy for the treatment of repetitive behaviour in neurodevelopmental disorders [21]. Commonly prescribed medications such as SSRIs (e.g., fluoxetine) have been shown to lack efficacy for repetitive behaviour in individuals with ASD as well as exhibit significant adverse effects [22,23]. Similarly, atypical antipsychotics (specifically, risperidone and aripiprazole), although there are some reports of efficacy on repetitive behaviour, have been approved by the Food and Drug Administration (FDA) only to treat irritability and not repetitive behaviour in persons with ASD. In addition, atypical antipsychotics are associated with significant weight gain and, potentially, metabolic syndrome with little evidence of efficacy for repetitive behaviour.

Stereotypies are present in a substantial proportion of the behavioural repertoires of people with ASD [24], typically beginning in early childhood and tending to persist, although they may decrease in frequency and duration [25] as the children grow older. Lovaas et al. found that children with ASD may have longer response latencies to sensory stimuli when engaged in stereotypic behaviours [26]. In children with ASD, stereotypies are often perceived as age-inappropriate in form, focus, context, duration or intensity.

As such, many clinicians consider them a relevant symptom to be targeted in behavioural interventions. Although repetitive and stereotyped behaviours traditionally have been considered to operate under sensory and automatic reinforcement contingencies, research has clarified that they may also be maintained by social or non-social positive and negative reinforcement. Indeed, it seems most appropriate to describe and categorize stereotypies in terms of their function, rather than their form [27].

This paper aims at mapping the literature on the main animal models of stereotypies and repetitive behaviour, in order to identify some neurobiological mechanism that can enhance our understanding of such alterations in persons with ASD.

## 2. Materials and Methods

Presenting an overview of a certain research area and identifying research gaps are the main goals of a structured mapping study [28]. This approach is organized into three main phases, namely planning, conduction, and reporting the review. In the planning phase, research questions are developed in order to define the review scope. The search string/s is/are also driven by the research questions. One constraint when defining a search string is that the result set is of manageable size, but still has the maximum possible coverage. Therefore, some additional synonyms and relevant terms that are most common for each attribute were selected and added to the search string.

In our mapping, the following questions were addressed: “are animal’s stereotypies abnormal?”; “does performing a stereotypy have a rewarding effect for animals?”; and “are there different kinds of animal stereotypies?”; “how are ASD stereotypies characterised?”; “what are the mechanisms underlying stereotypies that can bridge animal and ASD stereotypies?”

The article search was conducted in reference to the last 45 years using the search engines EMBASE with PubMED/Medline, PsycInfo, Medmatrix, NHS Evidence, Cochrane Library and Web of Science, during the second half of November 2015. The main search strings were: (“stereotyp*” AND “animal*”) and [(“stereotyp*” OR “repetitive behaviour” OR “restricted behaviour”) AND “autism” OR “autism spectrum disorder” OR (intellectual OR learning OR developmental AND disability OR disorder)]. Other strings including the same terms were used with reference to the following specific genetic syndromes including ASD or autistic features: Fragile X syndrome, Rett syndrome, Tuberous sclerosis, 22q11 deletion syndrome (velo-cardio-facial syndrome) and Cornelia de Lange syndrome.

The titles and abstracts of identified papers were screened. The articles which passed this filter were read in full and assessed in their capability to answer the reference questions. Titles and abstracts were independently checked by 2 researchers, both clinical psychologists working in the field of Intellectual Developmental Disorders and specifically trained in carrying out systematic reviews. Their agreement in relevance attribution was evaluated by concordance on the first 10% of identified material (K test) and found to be 99.6%. The full articles were read by the 2 psychologists and 2 psychiatrists, the latter also quite experienced in research and practice in Intellectual Developmental Disorders. Their agreement was checked by comparing the number of paragraphs per article that they had judged to address the mapping questions. Again, the concordance on the first 15 articles was very high (99.4%).

## 3. Results

A total of 570 articles matched the keywords. After titles were checked, 218 were selected. After abstracts were read, 166 were excluded, as they were not relevant to the mapping topic, and 2 were excluded because they were not in English.

After reading the remaining articles in full, 184 papers were considered relevant and included in a specific database (see Figure 1).

The selected papers were classified by sample size and study design, according to ARRIVE (Animal Research: Reporting of In Vivo Experiments) guidelines [29]. Most of the studies (61.1%) have a sample size ranging from 1 to 50 animals. The most common study design was Non Randomised Controlled Trial (45.8%). Result details are summarised in Table 1.

Key issues identified through the analysis of the literature were grouped in the following six areas for review: stereotypies seen in animals, stereotypies and repetitive behaviours in ASD, environmental deprivation, Central Nervous System (CNS) damage and repetitive behaviour, pharmacologically induced repetitive behaviour, and repetitive behaviour associated with specific inbred mouse strains.

### 3.1. Stereotypies in Animals

Stereotypies are a sub-type of ARB defined by the presence of perseverated actions that can be classified according to the nature of the action performed [30]. The term ‘abnormal’ refers to the being (1) significantly rarer and/or different with respect to a reference population or (2) related to a pathologic process [31]. The first meaning depends on our definition of “normality”. In this view, stereotypies may be considered “normal” in a captive population. On the contrary, by considering “normal” the condition of free-living animals, stereotypies have to be considered certainly “abnormal” [32,33], both for the context and the frequency of their performance [34].

Both ARB and stereotypies involve repetitive and ritualistic behaviours, but stereotypies are distinguished by their purposelessness, representing non-goal oriented actions or automatic and mechanical persistence. Usually Behavioural Perseveration (BP) and stereotypies are observed in animals kept in captivity, but they differ in some aspects.

BP is thought to result by innate cognitive inflexibility, deriving from an impaired neuroanatomical formation during development in captivity, while stereotypies develop in response to the environment.

Most animal studies about stereotypies have been focused therefore on captive animals to understand the right conditions for their welfare [35].

Stereotypies cannot be considered the product neither of natural selection (they are virtually absent in the wild condition), nor of selective breeding in captivity (they are not unique to domesticated species).

Anyway, a learnt behavioural pattern may be adaptive even if uncommon or unique. Stereotypies share some features with normal behavioural patterns [36]. They are invariant and resistant to change as some normal behaviours that are dependent on fixed environmental factors (i.e., grooming, drinking, etc.) [8,37,38,39,40]. Once developed, stereotypies gradually become independent from their original eliciting stimuli. Finally, a third important, and apparently ‘abnormal’, characteristic of stereotypy is that it has no obvious goal or function. One suggested explanation, concerning this last feature, is that the behaviour pattern is reinforcing in its own. Once developed, stereotypies are remarkably persistent, and this could indicate that the performance has some reinforcing value [41]. Then, certain stereotypies persist in spite of a significant energy cost. Finally, deterring an animal to perform a stereotypy is extremely difficult, also if alternative behaviour patterns are rewarded.

Moreover, some normal behavioural patterns may be elicited by situations of stress, conflict, frustration and, if repeated, become inflexible.

In a pragmatic view, only if the “cost” of stereotypies outweighs any benefit for the animal or the etiopathogenesis stands a pathological condition, it could be assumed that stereotypies are certainly “abnormal” in the maladaptive sense [42]. Following this perspective, the literature review was focused on the developmental process of stereotypies, usually named “escalation” [43], that consists of four distinct categories of behavioural change: ritualisation [42,44], emancipation [42], stabilisation [45] and cronicisation [42,46].

### 3.2. Stereotypies and Repetitive Behaviours in ASD

Repetitive sensory and motor behaviours can assume several forms in animals, depending on the species and the context in which they are observed. These can include excessive grooming, stereotyped pacing, backward somersaulting, rhythmic body movements, head twirling and excessive mouthing. These behaviours share important features with the ones observed in ASD, i.e., in being not only repetitive, but having little variation in the response form and no obvious function [15]. The use of animal models provides one approach for identifying the neural underpinnings of repetitive behaviours in clinical populations as chordates show the same basal ganglia basic circuitry, involved in behavioural control [47].

The occurrence of similar behaviours across species and in different neuropsychiatric and neurodevelopmental disorders had raised the question if they are caused by common mechanisms or have to be considered neurobiologically unique [13].

Identifying the neural networks of repetitive behaviours will help to shed light into the pathogenesis of neurodevelopmental disorders, stimulating new therapeutic initiatives.

In earlier studies, the neural basis of repetitive behaviour had been identified in the basal ganglia nuclei [3,48,49]. Research on repetitive behaviour on animal models was mostly based on motor stereotypies for three main reasons:Motor stereotypies are easier to model in animals;Higher-order repetitive behaviour in animals were thought to result from secondary neuropathological changes;It was believed that basal ganglia targeted only the cerebral cortex, involved in movement generation and control.

This initial point of view changed with the reformulation of basal ganglia theory by Alexander and collaborators [50], who proposed that basal ganglia have multiple and parallel circuits targeting not only the primary motor cortex, but also premotor and prefrontal areas. Five different circuits, all structured in a similar way, were defined (motor, oculomotor, dorsolateral prefrontal, lateral orbitofrontal and anterior cingulate circuit). Each circuit consists in two branches: the direct pathway—striato-nigral—and the indirect one—striato-pallidal.

The direct pathway positively modulates the activity of thalamus, whereas the indirect one exploits an inhibitory effect. This dual system serves as a fine activity tuner in different parts of the frontal cortex with effects on movement control, cognition and limbic functions [51].

Recently, it has become evident that different functions subserved by these circuits are interconnected, as information flows from higher order circuits to lower order ones [52].

This better understanding of the corticostriatal loops led to a re-evaluation of motor and non-motor repetitive behaviour animal models. The disruption of function within the basal ganglia or between the striatum and forebrain structures resulted in repetitive behaviours [53].

Many studies highlighted that the motor loop is involved in stereotypical motor behaviour, i.e., repeating identical movement without pursuing a goal [54]. Oculomotor perseveration has been described in individuals with schizophrenia and in animals (i.e., repeating eye-rolling in calves) [13].

The prefrontal loop has been associated with the repetition of goal-directed behaviour and is mainly implicated in the human repetitive behaviour [55]. Additionally, perseverations and impaired extinction learning have been associated with damage to the prefrontal cortex [13,56].

Finally, limbic loops are connected with motivational aspects of behaviour, as impulsive behaviour, response to reward and obsessive-compulsive behaviour.

Animal models relevant to restricted, repetitive behaviour in ASD generally fall into four classes: repetitive behaviours associated with restricted environments and experiences; repetitive behaviours determined by targeted insults to the CNS and gene mutations; repetitive behaviours induced by pharmacological agents; and repetitive behaviours in specific inbred mouse strains [5].

### 3.3. Environmental Deprivation

A large literature on repetitive behaviours investigated their connection with environmental restriction or social deprivation [18,55,57]. Deprivation-induced stereotypes are more prevalent in monkeys and apes than in lower mammals. This shows that humans may be particularly susceptible to such behaviour. In fact, as an example, adopted children from Romanian institutions have been reported to present negative effects of early social deprivation such as increased risk for self and hetero-directed aggressive behaviour, cognitive problems, and autistic-like features, including RRB [58,59], which continued even one year after adoption [60]. Deprivation for extended periods of time (six months or more) resulted in more severe and long-lasting consequences [61].

Stereotypies therefore have been thought to indicate that an animal’s environment is sub-optimal [32,34], and that the animal is suffering from a welfare problem [33].

ARB are commonly attended in animals housed in zoo, farm and laboratory environments, as well as in animals with an early social deprivation background. Stereotypies have been also associated with barren and restrictive conditions. This type of environment is thought to be sub-optimal [62], probably because there is a need for sensory stimulation per se [63]. Limitation of locomotor may cause the frustration of several motivations [64] and subsequently stereotypies, as shown in sows, [65], Rhesus Macaqus, Macacca Mulatta [66]. This view of repetitive behaviours stands on some features of stereotypies: (i) the context in which they develop, (ii) the behavioural patterns from which they arise, (iii) the factors that influence their development and (iv) the occurrence of self-damage. If an animal is motivated to perform a behaviour, but it cannot do so, the situations become frustrating [37,67] and usually result in redirected activities or, when the situation is chronic, in stereotypies [68].

Behavioural stereotypies in animals have been associated with sub-optimal environmental and also with psychological conditions. Therefore, they have been considered as stress expression [42]. Three basic needs, may cause stereotypies if not satisfied:Food intake [69];Locomotion [70];Social interactions [71].

According to Ridley’s hypothesis [72], environment shapes behavioural patterns and consequently a restricted situation may lead only to repetitive responses. Another interpretation is that confinement-induced stress may play only a mediating role in the development of repetitive behaviour, and stereotypies could be regarded as a coping strategy to reduce the arousal response to stressful events or environments [73].

Models of repetitive behaviour induced by environmental restriction may be of relevance to ASD. Actually, individuals with ASD suffer from deficits in a variety of domains, including social, emotional, motor and cognitive functioning. Thus, such children may be considered “functionally and environmentally restricted”.

It is worth noting that repetitive motor behaviour appears to be an invariant consequence of experiential deprivation or restriction in all tested species.

This hypothesis is supported by research on neurochemical effects of deprivation. For example, several studies have established that rats raised in isolation have significant forebrain catecholamine system dysregulation [74]. In other investigations, biochemical abnormalities in the striatal system, including changes in dopamine metabolism, have been directly demonstrated in deprived animals [75,76,77,78]. Furthermore, environmental enrichment determines structural and functional changes in striatal neurochemistry and prevents stereotypy development [79].

Environmentally deprived animals show cognitive abnormalities: poor extinguishing of learnt responses [35,80,81,82,83] and disinhibition of response selection [84].

These cognitive problems have been shown to be related to stereotyped behaviour.

In all considered species, the most stereotypic individuals also showed the most persistent, repetitive responses to different cognitive tasks [35,84,85]. These findings point to a link between deprivation-induced stereotypies and specific cognitive impairments, implying a common underlying route. Tanimura and colleagues [85] found in mice a strong link between stereotypies and cognitive rigidity mediated by corticostriatal circuitry (cognitive flexibility as measured by reversal learning). The specific process behind widespread motor and cognitive impairments, however, remains unknown. Furthermore, this type of repetitive behaviour does not appear to be an adaptive coping strategy [13], but the result of significant and possibly permanent abnormalities in brain development, as confirmed by the difficulty in treating this behaviour and consequent changes in striatal neurochemistry. Anyway, it is relevant to remember that the effects of environmental deprivation are not on an on/off scale, but are modulated by factors such as quality and duration of deprivation, genetic make-up and other individual characteristics [13].

While environmental restriction induces repetitive behaviour, environmental complexity is shown to ameliorate or prevent it. By providing extrinsic factors [86], or artificial substitutes to redirect the behaviour, stereotypies can be reduced. Factors that reduce or eliminate stereotypy are shown to be those that give the animal the opportunity to perform other behavioural patterns [87] or that decrease arousal [88]. Indeed, providing animals with more complex environments appears to be an effective means of attenuating repetitive behaviour [15,89].

Environmental enrichment has been associated with many Central Nervous System (CNS) effects, namely dendritic branching, spine pruning, synaptogenesis, angiogenesis, gliogenesis, gene expression, apoptosis and neurogenesis [35].

Nevertheless, this positive effect has been shown to be limited, particularly on long lasting stereotypies, even in combination with neurochemical treatment [49]. Other authors report that stereotypies may even be induced by overstimulating environments, which would represent an automatic defence mechanism against excessive stimuli and a way to preserve homeostasis and to reduce anxiety [90].

### 3.4. Central Nervous System (CNS) Damage and Repetitive Behaviour

Models of insult to the CNS (for example genetic mutations, viral exposure, lesions) associated with repetitive behaviours, are particularly intriguing because they promise clues to etiology and pathophysiology [58].

Studying behaviour of transgenic animals (often mice), can enhance our understanding of the role of genes in the development of abnormal behaviour.

Two main branches of research concerning CNS damage and repetitive behaviours can be identified.

#### 3.4.1. Non Genomic Factors

Studies investigating effects of brain lesions on repetitive behaviour indicate a central role of the striatum [5,58], and its reciprocal connections with some main areas of the medial temporal lobe [91]. According to Gao and Singer [92], the cortical-striatal-thalamo-cortical processing loop, which is crucial for movement initiation, continuation, and termination, would be altered in ASD and other psychiatric disorders including complex motor stereotypies or repetitive behaviours, such as obsessive–compulsive disorder, or the Tourette syndrome. Changes in stereotyped behaviour have also been reported after lesions to the non-striatal structures within the medial temporal lobe structures (hippocampus, amygdala). The brain circuit is believed to play a role in the expression of motor stereotypies.

According to Bauman and collaborators, early limbic system lesions affect the development of other brain regions as well, such as the medial prefrontal cortex, which is involved in the regulation of striatal dopamine activity [91].

Some studies have tried to investigate the effects of mechanically induced and localized CNS lesions on the development of repetitive behaviours. An increase in stereotyped behaviours has been associated with lesions in the substantia nigra pars reticulata (SNpr), presumably due to the disinhibition of dopamine neurons in the substantia nigra pars compacta (SNpc) [93].

From these studies, it appears that the timing of lesions is crucial for the emergence of behavioural abnormalities [91,94,95,96]. Findings of other studies seem to suggest the presence of specific windows in the development of repetitive behaviours [97].

Other animal models have examined the role of prenatal risk factors in the etiology of ASD. Some of these models have been generated based on the observation that prenatal exposure to teratogenic agents increases the risk of ASD.

Exposure to valproic acid (VPA), an antiepileptic drug, on embryonic day 12.5 in rats produces neuroanatomical abnormalities similar to those reported in autistic individuals, but also long term disturbances in postnatal behaviour, as increased time spent in stereotypic activity [57,98,99]. The stereotypies expressed by the VPA-treated rats can be modulated by environmental perturbations, such that housing in an enriched environment results in an attenuation, according to the results reported above. Moreover, the offspring of women taking VPA for mental illness or epilepsy during early pregnancy are at an elevated risk for ASD [99,100,101].

The key mechanism underlying the effects of maternal VPA on foetal brain development are still unclear, since the wide range of VPA effects, including altered gene expression, cell death and immune dysregulation. Interestingly maternal VPA exposure leads to reduced expression of neuroligin (NLGN) [102], an ASD candidate gene.

#### 3.4.2. Genomic Factors

Concurring with pharmacological studies, genetic models have implicated the dopamine system in repetitive behaviour. These models mainly include the dopamine transporter (DAT) and dopamine receptor D3 (DRD3) knockout mice and the dopamine receptor D1 (D1) mutant mouse. Such models may be particularly informative on the spontaneous development of repetitive behaviour [13].

The effects of knocking out the DAT-gene are shown to be dual. First, it results in increased dopamine levels at level of neostriatum up to 170% [103]. The hyperdopaminergic DAT-null mice therefore display behaviour known as “super-stereotypy”, which is characterised by extremely strong and rigid manifestations of complex and fixed patterns of action.

Second, the DAT gene knocking out leads to an imbalance between the dopamine and serotonin systems in the basal ganglia [104].

Unlike the intense and different behavioural effects observed in a DAT-null mouse, the effects of knocking out the dopamine D3 (DRD3) receptor gene are more specific and lead to defined changes in behaviour. Joseph and coll. pointed out an increase in spontaneous stereotypic behaviour of DRD3-knockout mice comparing to the wild type [105].

A potential pitfall with such translational models may be that modifications affect the entire organism, on one hand by generating non tissue-specific effects, on the other one giving rise to possible compensatory mechanisms.

Gene manipulation targeted to specific brain regions may lead to further understanding of the modulatory effects of the involved genes. Campbell and colleagues investigated behavioural abnormalities in transgenic mice following the pharmacological potentiation of regional subsets of dopamine D1 neurons (in cortical and limbic regions) [106]. Treated mice exhibited bouts of persistence and repetition of all behaviours, including leaping and non-aggressive biting of siblings during grooming.

This result suggests that boosting regional activity within corticostriatal loops by genetically altering particular parts of the dopamine system can generate compulsive behaviour in mice [13].

Beside dopamine system genes, the number of genes that may potentially affect abnormal repetitive behaviour is very large. The main ones are summarised in Table 2.

### 3.5. Pharmacologically Induced Repetitive Behaviour

In animal models, stereotypies and repetitive behaviours can be produced by drugs that stimulate dopamine release, dopamine reuptake inhibitors, and direct dopamine receptor agonists [13,121]. The observation that older adults with intellectual disability who have high pervasive stereotypic movements exhibit lower rates of eye-blinking and greater variance in eye blinking intervals, is aligned with these findings. In view of the fact that eye blinking rate has been found to directly correlate with dopamine function, these results suggest that stereotypies are connected to dopaminergic dysfunction [122].

In this section, we will consider drugs induced stereotypies, which manipulate the brain circuitry more directly. Furthermore, pharmacological manipulations can be applied in young animals to assess their impairment in the development of repetitive behaviour, which is more relevant with respect to ASD.

Pharmacological studies have provided a great amount of what we know about the relevant neurobiological circuitry and a number of the drugs (for example cocaine, amphetamine) used to induce repetitive behaviours in animals that can also induce them in humans [57].

Early experiments highlighted the importance of basal ganglia in mediating the induction of repetitive behaviour by such drugs. The basal ganglia system is modulated by different endogenous peptides. The main neurotransmitters in striatum, pallidum and thalamus are GABA and glutamate. Corticostriatal circuits are further modulated by dopamine, opiates (dynorphin, enkephalin), serotonin and several other neurotransmitters [13,35].

For example, increased levels of extracellular dopamine in the dorsal striatum were associated with decreased levels of acetylcholine (ACh) release. This imbalance seems to be related to the severity of motor stereotypies [121]. Studies investigating the role of neurotransmitters in repetitive behaviour are faced with a number of complications: firstly, these systems do not function in isolation, but are interactive; moreover, the manipulation of a single system may influence the other ones. Secondly, when exogenous pharmacological agents are employed to affect these systems, it is relevant the way of administration: the effects of direct injection into a brain region may be very different from the effects of oral, subcutaneous or intravenous administration. Thirdly, the effects of exogenous agents are usually dose-dependent, complicating the generalisation of findings regarding drug-induced behaviour [123].

A summary of findings regarding the main agents involved in stereotypies is reported in Table 3.

### 3.6. Repetitive Behaviour in Specific Inbred Mouse Strains

Inbred strains of mice have recently become the most frequently employed model for studying human brain disorders [5]. Identifying an inbred strain exhibiting repetitive behaviour without requiring a specific perturbation (lesion, drug or genetic mutation) is believed to be of significant importance to this field.

At least two inbred strains appear to be good models for ASD.

The BTBR T+tf/J (BTBR) inbred mouse strain displays several physical and behavioural abnormalities, constituting a reliable face validity for modelling ASD, because it exhibits a number of autistic-like traits, including repetitive behaviour in the form of elevated levels of self-grooming [145,146] and also cognitive inflexibility.

Complex or higher order repetitive behaviour (rituals, insistence on sameness, restricted interests) in individuals with ASD reflects a cognitive rigidity or inflexible adherence to routines and rituals [5,57].

To address the resistance to change/insistence on sameness behavioural domain, Amodeo and collaborators employed a spatial reversal learning task with BTBR mice [147]. Compared to C57BL/6 mice, BTBR mice performed similarly to controls in acquiring the spatial discrimination, but were impaired on reversal learning. This impairment was only observed when feedback for a correct choice was decreased to an 80% probability. BTBR mice also display inflexibility in the exploration of a hole-board and more patterned sequences in sequential investigations of a novel object, suggesting that this strain demonstrates both cognitive inflexibility and stereotypic motor behaviours [145,148].

Recent work has shown that the degree of restricted, repetitive behaviour in individuals with ASD correlates positively with deficits on executive function tasks that index cognitive flexibility [149].

Cognitive flexibility, or resistance to change, can be assessed in animals using a variety of tasks that range in complexity from response extinction to reversal learning to intra- and extra-dimensional set shifting [150]. Work conducted with several different species has demonstrated that motor stereotypies are inversely correlated with measures of cognitive flexibility. For example, in bank voles and bears, extinction learning was significantly inversely correlated with the amount of stereotypy [80,81].

Similarly, Orange wing Amazon parrots were assessed for stereotypy and performance on a variation of a gambling task, which indexed the tendency to repeat responses or perseverate.

Animals with higher stereotypy scores exhibited greater sequential dependency in their responses on this task [80]. Lewis et al. examined the performance of deer mice in a procedural learning task that involved learning to turn down the right or left arm of a T-maze for reinforcement [57]. After acquisition, learning was reversed. Results indicated that high levels of stereotypy in deer mice were associated with deficits in reversal learning in the T-maze. In a hole board task, BTBR mice display inflexibility in exploratory behaviour and fail to shift exploration away from a familiar bedding stimulus to a palatable food odour [148].

Such relationships between cognitive rigidity (deficits in set shifting, extinction, and reversal learning) and motor stereotypy can be understood given the common mediation by cortical-basal ganglia pathways.

Alterations in this circuitry could well impair the ability to inhibit pre-potent responding, the ability to orient to novel events and the ability to generate flexible patterns of behaviour.

Pierce and Courchesne assessed groups of B6 and BTBR mice for the frequency of repetitive contacts with novel objects: the object preferences and more invariant patterning of object exploration in this study provide striking parallels to the reduced toy exploration noted in children with ASD diagnoses relating to stereotyped patterns of behaviour, interests and activities [151]. These include an “encompassing preoccupation with one or more stereotyped and restricted patterns of interest…inflexible adherence to specific, nonfunctional routines or rituals… and a persistent preoccupation with parts of objects [152].” These findings indicate that BTBR mice show cognitive aspects of stereotypy, in addition to display of motor sequences, such as those involved in grooming and bar-biting.

Investigation of repetitive stereotyped behaviours suggests that these traits may be reflective of functional homologies rather than superficial parallels in restricted interests and behaviours seen in the clinical population and within the inbred model. Such phenomena might be helpful in detecting clinically-relevant endophenotypes that are collectively present in the BTBR strain, which in turn can be applied in other candidate strains and mutants for the ultimate aim of unravelling the complex psychopathology of ASD.

Another inbred mouse strain that seems to be promising for furthering our understanding of the neurobiology of repetitive behaviour is the C58 strain. Ryan and coll. reported repetitive hindlimb jumping and persistent backflipping in these mice [153,154].

Inbred C58/J mice also demonstrated stereotypic hyperactivity and back-flipping, in addition to lack of sociability and poor learning acquisition [5].

Remarkably, these repetitive motor responses in C58/J mice may be reduced through environmental enrichment.

These observations have been confirmed beyond [155], showing that, compared to C57BL/6 mice, C58 mice exhibited high rates of spontaneous hindlimb jumping and backward somersaulting. Authors also showed that six weeks of environmental enrichment following weaning substantially reduced repetitive behaviour. C58 mice did not exhibit increased marble burying, nor did they display reduced exploratory behaviour in the hole-board task. Further investigation of cognitive inflexibility in this strain, therefore, will be important in determining the utility of this model for modelling resistance to change/insistence on sameness.

### 3.7. Different Kinds of Stereotypies

As above mentioned, RRB represent one of two diagnostic domains for ASD, referring to a broad range of behavioural patterns that include stereotypies, repeated self-injurious behaviour, repetitive manipulation of objects, rituals and routines, and insistence on sameness. As a result of the present literature mapping of animal research, RRB can be distinguished in “lower-order” motor actions and “higher-order” behaviours. The former are characterised mainly by repetition of movements and include stereotypies, repeated self-injury, and repetitive manipulation of objects, while the latter are characterised mainly by rigidity or inflexibility, and more complex behaviours, and include rituals and routines, and insistence on sameness [6,7,156,157]. Transposing this distinction to stereotypies of persons with ASD and corroborating it with their own clinical experience, the authors of the present paper propose to distinguish among stereotypies a subgroup characterised by higher complexity, articulation, association to individual history, modulation by environmental factors, and finalisation. This group is here suggested to be named as ‘autotypies’, to be distinguished from other stereotypies, which are simple behaviours or acts, rigid, independent from its original eliciting stimulus, and often reinforcing in its own.

## 4. Discussion

Stereotypies represent a wide range of abnormal and repetitive behaviours and core deficits in ASD [2,12]. The reviewed literature highlights how animal models could be useful in reproducing such behaviours, connected to different human disorders, and particularly linking them to ASD.

The aim of this study is revealing the etiopathological and neurobiological basis of the disorder and, in turn, the different endophenotypes associated. This may lead to new pharmacological treatments [13,158].

Early studies had already highlighted the role of basal ganglia circuitry in repetitive behaviours [48], but later the functional and structural anatomy of different cortico-striatal loops was further underlined [50]. These findings pointed out how selective damage to each loop caused different type of stereotypies, according to the cortical area targeted.

Besides, as research was proceeding, a new line of investigation emerged and is nowadays active, i.e., the attempt to model insistence to sameness of ASD in mice model. Following this new line of investigation, insistence on sameness emerged as connected with the level of motor and behavioural stereotypies. Both kinds of stereotypy actually depend on the basal ganglia circuitry, which is also connected with different cerebral areas, upon which restricted interests, mannerisms and persistent preoccupation with parts of objects in ASD depend [159,160].

However, all the papers reviewed stress that there are a number of difficulties in developing animal models with features of ASD [161]. First, the disorder is currently defined by a set of core behavioural abnormalities rather than by objective biomarkers. Second, ASD may actually represent a set of behaviourally distinct disorders, with different causes and pathogenesis [162]. Use of the broader definition, as ASD, further complicates this issue, also in diagnosis and detecting especially people without associated intellectual disability [163]. Third, the genetics of ASD are complex, encompassing numerous candidate genes, copy number variations, and monogenic, syndromic disorders, also with autistic symptoms [164,165]. Lastly, in animals there is currently no pathognomic feature of ASD that can be clearly used to distinguish an “autistic” from a “schizophrenic mouse”, and in humans the relationship between psychosis and ASD is complex [166,167,168].

Even with reference to RRB, animal models that can represent salient features of human neurodevelopmental disorders are limited. In fact, most of these models centre around sensorimotor, or lower-order, RRB, although there are select models that display analogues for higher-order RRB, such as reversal learning deficits [14].

Other issues are related to the primary and secondary nature of stereotypies with respect to various environmental factors and psychological distress. Most of reviewed studies indicate stereotypies to be subordinate to current or past aversive environment or events, but several hypotheses have also been advanced on an inverse order, as well as a reinforcing, coping, and rewarding function of stereotypy [42,169,170].

Though much of current knowledge on the neurobiological basis of stereotypies comes from studies of drug-induced RRB, very little of the work we have reviewed addressed the identification of specific potential therapeutic targets of pharmacological treatment of stereotypies using animal models. This is a critical need in the field as there are few, if any, pharmacological interventions for the treatment of RRB in ASD with established efficacy.

Results of the present review may be limited by the procedure followed and the breadth and complexity of the scope. A literature mapping varies from a systematic review in the breadth of the topic area and questions, and the limits of data extracted [28]. However, some process for performing systematic mapping studies, such quality criteria when evaluating the identified articles, have been considered in addition while carrying out the present work.

Basing on our results, we suggest for future research in the field to shift the focus from complex syndrome (ASD) studies to inter-species trait studies, which may enable a more precise definition of cross-species stereotypic behaviours and facilitate the identification of underlying biological substrates. Detailed phenotyping and consensus in the definitions applied is indispensable to systematic research efforts investigating repetitive behaviour across species and clinical conditions. Future studies should also address more generally the degree of usefulness of applying a phylogenetic approach to this symptomatic dimension of ASD through new phylogenetically based analytical methods. For accurate inference, statistical studies of comparative data should assume some model of character evolution and taxa used in comparative analysis should be selected basing on their phylogenetic affinities [171].

## 5. Conclusions

Literature mapping confirmed that phylogenic approach and animal models may help to improve understanding and differentiation of stereotypies in ASD. Some repetitive behaviours appear to be associated with restricted environments and experiences, pharmacological agents and common genomic and anatomical factors across species, mainly alterations of basal ganglia circuitry.

Furthermore, a distinction between stereotypies and autotypies should be considered, with the latter representing a new subgroup of RRBs characterised by higher complexity, articulation, association to individual history, modulation by environmental factors, and finalisation than stereotypies.

Therefore, knowledge derived from the phylogenic approach and from studies on animal models may contribute to our understanding of the neurobiological mechanisms underlying RRBs and consequently support clinical issues related to stereotypies in persons with ASD, as well as provide new in-sights in classification, pathogenesis, and management. However, future studies must validate and refine this utility and breadth by defining cross-species stereotypic behaviours more precisely and using modern phylogenetically based analytical methods.

## Figures and Tables

**Figure 1 brainsci-11-00762-f001:**
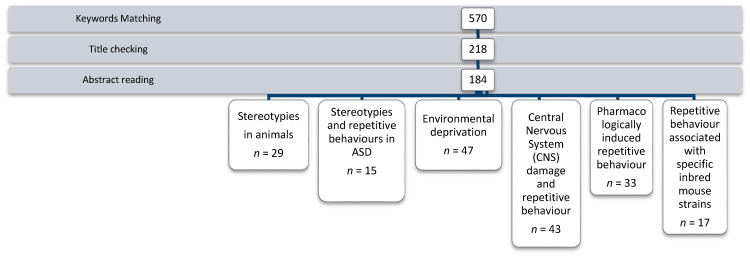
Paper selection process of our systematic mapping.

**Table 1 brainsci-11-00762-t001:** Level of evidence of studies considered for the literature mapping, in accordance to ARRIVE guidelines.

		*n*	%	
**Sample Size**	Not specified		16.3	
			83.7	
	Specified	1–50	51.2	61.1
		51–100	16.3	19.4
		101–150	6.9	8.3
		151–200	2.3	2.7
		200+	6.9	8.3
**Study Design**	RCTS (Randomised controlled trial)		22.9	
		Blind	27.2	
		Not blind	72.7	
	NRS (Non-randomised controlled trial)		45.8	
		Blind	45.4	
		Not blind	54.5	
	Cohort Study		20.8	

**Table 2 brainsci-11-00762-t002:** Genes that may potentially affect abnormal repetitive behaviour.

	Findings	References
GABA A-receptor beta-3 (GABRB3)	ASD, together with Prader-Willi and Angelman syndromes, has been linked to changes in a specific region (q11–13) of chromosome 15 carrying the GABRB3 gene	[107] DeLorey TM et al., 2008
Hoxb8	Repetitive behaviour can also be observed in mice with perturbations to the Hoxb8 gene, which display excessive grooming up to wound infliction	[108] Greer JM and Capecchi MR, 2002
Serotonin receptor 2C (HTR2C or 5-HT2c)	The 5-HT2c knockout mouse shows intensified and stereotyped chewing and reduced habituation of responses	[109] Chou-Green JM et al., 2003
	HTR2C has been linked to ASD and Obsessive Compulsive Disorder	[110] Veenstra-VanderWeele J et al., 2000
SAPAP3 (disks large-associated protein-3 gene DAP-3 or SAP90/PSD- 95-associated protein 3 or SAPAP3)	SAPAP3 produced a mouse model of reduced cortico-striatal synaptic transmission and glutamate receptor function and excessive self-grooming behaviour	[111] Welch JM et al., 2007
SHANK gene family (SHANK1, SHANK2, and SHANK3)	SHANK1 deletion has been identified in a small number of males with higher-functioning ASD	[112] Sato D et al. 2012
	SHANK2 and SHANK3 mutations have been found in some patients with ASD and intellectual disability	[113] Berkel S et al., 2010
	Disruption of the SHANK3 gene in mice results in functional deficits in glutamatergic synapses and autistic-like behaviours, which includes repetitive behaviour in the form of increased grooming, sniffing and object manipulation.	[114] Schmeisser MJ et al., 2012
Neuroligin and neurexin genes	NL1-null mice groomed for twice the amount of time with respect wild-type animals	[115] Blundell J et al., 2010
	Generation of neurexin1α deficient mice revealed behavioural changes, including increased grooming and impaired nest-building behaviour, although no obvious deficits in social behaviour or learning	[116] Etherton MR et al., 2009
M2 muscarine acetylcholine receptor	M2 muscarine acetylcholine receptor knockout mice show marked impairment in set-shifting in the Barnes circular maze task, with an increased perseverative behaviour	[117] Dallaire JA et al., 2011
NRG1	Neonatal mice treatments with recombinant eNRG1 protein and T1-NRG1 similarly enhance methamphetamine-triggered stereotypic movements	[118] Kato et al., 2015
**Grin1**	Grin1 knockdown mice have reduced NMDA receptor function and demonstrate spontaneous motor stereotypy, including over-grooming and self-injury	[119,120] Moy et al., 2008; Gandal et al., 2012

Legend: GABRB3: Gamma-aminobutyric acid receptor subunit beta-3; Hox-B8: Homeobox protein B8; HTR2C: Hydroxytryptamine Receptor 2C; 5-HT2c: 5-Hydroxytryptamine Receptor 2C; SAPAP3: SAP90/PSD-95-associated protein 3; DAP-3: Disks large-associated protein 3; SAP90: Synapse-associated protein 90; PSD- 95 associated protein 3: postsynaptic density 95-associated protein 3; SHANK: SH3 (SRC Homology 3) and multiple ankyrin repeat domains; NRG1: Neuregulin 1; Grin1: glutamate ionotropic receptor NMDA type subunit 1; NMDA: N-methyl-D-aspartate.

**Table 3 brainsci-11-00762-t003:** Pharmacologically induced repetitive behaviour.

Neurotransmitter	Findings	References
**GABA**	Injection of Gamma aminobutyric acid (GABA) agonists in the substantia nigra pars reticulata (SNpr) induces stereotypies in rats	[124] Scheel-Kruger et al., 1980
	The administration of GABA-agonists or antagonists to the frontal cortex in rats respectively attenuates or exacerbates stereotypies	[125] Karler et al., 1995
	Distinct behavioural effects following GABAergic drug administration, affected by topographical variations relative to the site of injection or dose differences	[13] Langen et al., 2011
	Microinjection of GABA antagonist (bicuculline) into the limbic portion of the external Globus Pallidus induces stereotypies in monkeys	[126] Grabli et al., 2004
	Lower levels of GABA observed in the anterior cingulate cortex are predictors of symptom severity	[127] Harris et al., 2016
**Glutammate**	Glutamate receptor agonists, as N-methyl-D-aspartate (NMDA) agonists, can induce stereotypic behaviour, whereas administration of an NMDA-receptor antagonist into striatum can attenuate drug-induced stereotypy	[128] Bedingfield et al., 1997
	Involvement of increased levels of glutamate and aspartate in the striatum in the mediation of stereotypic behaviour in mouse models	[129] Presti et al., 2004
**Dopamine**	Administration of apomorphine, a dopamine agonist, was shown to activate dopamine receptors in the neostriatum, resulting in compulsive gnawing behaviour	[54] Saka et al., 2004; [13] Langen et al., 2011
	Striatal dopamine may influence the balance between direct and indirect pathways, affecting the global level of basal ganglia output	[130] Groenewegen et al., 2003
	Dopaminergic drugs may modulate the prevalence of stereotypy by stimulating the direct pathway and inhibiting the indirect one	[35] Mason and Rushen, 2006
	The activation of striatal D1 receptors by dopamine projections from substantia nigra pars compacta (SNpc) results in the amplification of excitatory corticostriatal input and subsequently increases GABA-ergic inhibition of the SNpr and the globus pallidus. Conversely, blocking dopamine D1 receptors suppresses the direct pathway, and decreases feedback to the cortex, resulting in less stereotypic behaviour.	[131] Joel and Doljansky, 2003; [132] Presti, 2003
	Dopamine agonists acting on D2 receptors (e.g., amphetamine, apomorphine) suppress the indirect pathway and disinhibit behaviour	[5] Bechard et al., 2012
	Dopamine antagonists reduce stereotypies by blocking dopamine D2 receptors	[133] Kjaer et al., 2004
	Raclopride, a dopamine D2 antagonist, may determine the arrest of stereotypies when injected in the prefrontal dorsal striatum while it has no effect when injected in the sensorimotor area of the dorsal striatum.	[121] Aliane et al., 2011
	Administration of the GABA antagonist bicuculline into the frontal cortex enhances the motor stimulatory effects of amphetamine	[134] Kiyatkin et al., 1999
	Amphetamine-induced stereotypy can be attenuated via intracortical infusion of dopamine	[57] Lewis et al., 2007
**Serotonine**	Pharmacological stimulation of postsynaptic serotonin receptors in rodents leads to complex behavioural symptoms including stereotyped and repetitive behaviour	[135] Curzon, 1990
	Spontaneous stereotypic behaviour is associated with hypoactivity in serotonin pathways	[136] Korff et al., 2008
	Stereotypy-reducing effects of citalopram in bank voles	[137] Schoenecker and Heller, 2003
	Higher serotonin release and/or overactivity of serotonin receptors in the development of repetitive behaviour	[13] Langen et al., 2011
	Serotonin may affect the development of stereotypies by modulating the dopamine system	[138] Schoenecker and Heller, 2001
	Dopamine-induced motor stereotypies can be alleviated by drugs that act on serotonin receptors	[139] Elliott et al., 1990
	Motor stereotypies in rats given large doses of amphetamine have been shown to be dependent on serotonin release	[140] Lees et al., 1979
	Stereotypy can be reduced by injection of serotonergic antagonists into the Sub-Thalamic Nucleus	[141] Boris et al., 2007
**Oxytocin and vasopressine**	Functional alterations in these systems may contribute to social deficits in ASD and to repetitive behaviours	[142] Insel, 2010; [143] Ross and Young, 2009
	Genetic variations in Oxytocin (OT) receptor and vasopressin receptor 1A (V1aR) can be associated with ASD	[101] Patterson, 2011
**Histamine**	Acute dose of ciproxifan, an histamine H3 receptor (H3R) antagonist, is able to attenuate some stereotypies present in the animal model of ASD induced by valproic acid	[144] Baronio et al., 2015

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
