# Peer review of "Stereotypies in the Autism Spectrum Disorder: Can We Rely on an Ethological Model?"

_brainsci, 2021, doi:10.3390/brainsci11060762_

Round 1
Reviewer 1 Report
The manuscript entitled " Stereotypies in the autism spectrum disorder: can we rely on an ethological model?" is very organised and well written. The manuscript explains sterotypes in ASD in a good manner and has in-depth knowledge in their area of research and it reflects in their manuscript.
However, there are a few minor changes that can be made to enhance the quality of the manuscript.
Page 2- line 69: Could you please add a reference for this sentence.
Page 3- line 127: Since, the keyword search is more than 4 years old, how could you justify this with the present day results?
Table 2- Seems irrelevant.
Tabke 3- The genes mentioned is very limited and could include a couple more.
Tabke 4- Histamine is one of the important neurotransmitters but is not included. Could you please justify that?
Reviewer 2 Report
The paper takes into consideration an important aspect of autism starting from animal studies. the literature is organized around six areas which then constitute the six points of the results section.
The work is interesting but needs to be revised and edited with greater precision. For example on page 4 authors say that there are three areas when in reality they are six. Among the areas a seventh area is added ‘other kind of stereotypies’ which in fact is not an area because it has no literature and seems to be a theme of ​​interest which is more appropriate to use in the introduction and in discussion rather than in the results section.
Then there are several other inaccuracies in the work.
In tab1 it is advisable to put the number of papers directly in the area box.
In table 2, the percentages must be reported exactly: for example, the sample size is specified in 83.7% of papers and therefore the percentages must be reported relative to this percentage. The same as far as study design is regarded.
Chapter 3.2 must show the same title of the area identified.
3.5 and 3.6 have to be reported as subheadings of 3.4
In the introduction a reference to stereotypies in typical development is missing.
A more important point that deserve major revision is the reorganization of discussion around the most important finding in the six area and there importance for human studies. Discussion could be a bit longer and two sections about limitation (that could begin with the last sentence in discussion) and future direction (where could be discussed the phylogenetic approach that at the moment I consider inappropriate for this type of revision)
